# Exploring the Biological Impact of β-TCP Surface Polarization on Osteoblast and Osteoclast Activity

**DOI:** 10.3390/ijms26010141

**Published:** 2024-12-27

**Authors:** Jingpu Zheng, Kosuke Nozaki, Kazuaki Hashimoto, Kimihiro Yamashita, Noriyuki Wakabayashi

**Affiliations:** 1Department of Advanced Prosthodontics, Graduate School of Medical and Dental Sciences, Institute of Science Tokyo, Yushima, Tokyo 1138549, Japan; zhengjingpu01@126.com (J.Z.); yama-bcr@tmd.ac.jp (K.Y.); wakabayashi.rpro@tmd.ac.jp (N.W.); 2Department of Applied Chemistry, Faculty of Engineering, Chiba Institute of Technology, Narashino 2750016, Japan; kazuaki.hashimoto@it-chiba.ac.jp; 3Advanced Comprehensive Research Organization, Teikyo University, Itabashi, Tokyo 1730003, Japan

**Keywords:** β-tricalcium phosphate (β-TCP), electrical polarization, surface charge, osteogenesis, osteoclast

## Abstract

β-tricalcium phosphate (β-TCP) is a widely utilized resorbable bone graft material, whose surface charge can be modified by electrical polarization. However, the specific effects of such a charge modification on osteoblast and osteoclast functions remain insufficiently studied. In this work, electrically polarized β-TCP with a high surface charge density was synthesized and evaluated in vitro in terms of its physicochemical properties and biological activity. Polarization was performed to achieve a high surface charge density, which was quantified using a thermally stimulated depolarization current. The proliferation and differentiation of MC3T3-E1 osteoblast-like cells were assessed via WST-8 and alkaline phosphatase assays. Tartrate-resistant acid phosphatase (TRAP) activity and a resorption pit assay were used to evaluate the impact of surface charge on RAW264.7 osteoclast-like cell activity. Polarized β-TCP exhibited a surface charge of 1.3 mC cm^−2^. Electrically polarized surfaces significantly enhanced osteoblast proliferation and differentiation. TRAP activity assays demonstrated effective osteoclast differentiation of RAW264.7 cells, with enhanced activity observed on charged surfaces. Resorption pit assays further revealed improved osteoclast resorption capacity on β-TCP surfaces with a polarized charge. These findings indicate that β-TCP with a highly dense surface charge promotes osteoblast proliferation and differentiation, as well as osteoclast activity and resorption capacity.

## 1. Introduction

Bone graft materials play a crucial role in the fields of clinical and bone tissue engineering. These materials are surgically implanted to repair bone defects caused by trauma, fractures, tumor resection, skeletal deformities, or other conditions, thereby filling and reconstructing the defective area and promoting bone tissue regeneration and healing [1,2]. According to their sources, bone graft materials can be categorized as autografts, allografts, or synthetic bone materials. Autografts can lead to complications related to bone harvesting, and are further limited by insufficient quantities of bone available for harvesting. Compared with autografts, allografts do not have the functions of osteogenesis and bone induction and carry potential risks of viral and other pathogen infections [3,4]. Implant materials are required to have excellent mechanical properties, good biocompatibility, and superior corrosion and wear resistance. Typically, polymers, metals, ceramics, and their composites are used as implant biomaterials [5,6]. Owing to their similarity to the mineral components of bone, significant biocompatibility, bioactivity, biodegradability, and excellent mechanical properties, bioceramics have become a research hotspot in bone transplantation research [7,8,9]. Various modifications have been made to bioceramics to enhance interactions with natural tissues, promote desired cellular responses, and facilitate regeneration. These include the incorporation of bioactive molecules, such as magnesium, lithium, and strontium ions, as well as fibrin and vascular endothelial growth factor (VEGF) [10,11,12]. Studies have demonstrated the potential of hydroxyapatite nanoparticles in enhancing bone regeneration, improving osteoconductivity, and facilitating interactions with biological systems [13,14,15,16]. Moreover, the implantation of mesenchymal stem cells (MSCs) to promote tissue regeneration and the effects of the microstructure and surface morphology on cellular behavior have been explored. These strategies not only support but also actively play a role in the healing and regeneration processes [17,18,19]. Over the past 50 years, various types of bioactive ceramics have been developed by modifying the ion composition, ratios, ultrastructure, and mechanisms of action. Several bioceramics are now commercially available and have received FDA approval [20].

β-tricalcium phosphate (β-TCP) is a bioceramic material with an inorganic composition similar to that of bone matrix, exhibiting excellent biocompatibility and bioactivity [7]. TCP can improve the proliferation of bone precursor cells, such as osteoblasts and bone marrow stromal cells [21]. The nanoporous structure of β-TCP provides favorable biomineralization and cell adhesion characteristics [22]. Additionally, β-TCP, owing to its controllable ceramic structure, is biodegradable. In clinical trials conducted by Chung et al., β-TCP scaffolds showed a resorption of approximately 55% within 12 months [23]. Additional clinical studies have indicated that β-TCP demonstrates equivalent healing rates, clinical success rates, and patient outcomes compared with autografts, with lower rates of treatment failure, pain, and complications [24]. β-TCP implantation in vivo leads to its absorption and bone tissue replacement, exerting a significant influence on the bone formation and remodeling processes. This property has become a hot topic of research on bone graft materials in recent years [25,26].

A surface charge in ceramics can be achieved by manipulating dipole moments at high temperatures and stabilizing them at room temperature [27]. This polarization technique is mature, stable, and suitable for industrial-scale production [28]. Recent studies have demonstrated the application potential of this technology in the field of biomaterials. Polarization has been found to enhance the surface hydrophilicity and early bone integration of biomaterials [29,30,31]. The polarized form of another commonly used biomaterial, hydroxyapatite (HAp), has been reported to promote bone formation [32,33,34]. However, the polarization of β-TCP has rarely been studied. According to Wang et al., under the effect of polarization, the ceramic particles in β-TCP accumulate a charge 10^2^ times greater than those in HAp (10 μC cm^−2^) [35,36]. The use of polarized β-TCP particles can reportedly promote bone formation in the maxilla of dogs [37]. However, a polarized β-TCP scaffold suitable for clinical applications remains unexplored. Given the outstanding biocompatibility and degradability of β-TCP, the clinical application prospects of polarized β-TCP are expected to be extensive if developed.

This study was primarily focused on developing a form of polarized β-TCP suitable for clinical applications and investigating its physicochemical and biological properties in vitro.

## 2. Results

### 2.1. Electrical Polarization of β-TCP Samples

The thermally stimulated depolarization current (TSDC) patterns for both polarized and unpolarized β-TCP samples are presented in Figure 1. The TSDC curve for the polarized samples exhibits a single peak at approximately 570 °C. The depolarization current density begins to increase at 320 °C, reaching its peak at 570 °C. Based on the TSDC data, the charge density (*Q*) calculated for the polarized β-TCP was determined to be 1.3 mC cm^−2^. In contrast, the TSDC curve for the unpolarized β-TCP samples shows no variation with increasing or decreasing temperature, forming a straight line near the position where the current density approaches zero.

### 2.2. Characterization of Polarized β-TCP Samples

Figure 2 shows the X-ray diffraction (XRD) patterns of β-TCP samples before and after polarization. The XRD pattern of the polarized sample aligns with those of β-TCP (ICDD: 00-055-0898), confirming the absence of impurities in the β-TCP samples.

Figure 3 presents the Fourier-transform infrared (FT-IR) spectra of the β-TCP samples, both before and after electrical polarization. In the FT-IR spectrum of β-TCP, peaks corresponding to vibrations of the PO_4_ groups were observed at 422, 552, 592, 606, 943, 969, 1023, 1041, 1077, and 1120 cm^−1^ [38]. Notably, there was no evidence of the characteristic pyrophosphate (P_2_O_7_) band at 870 cm^−1^ or the distinctive absorption at 630 cm^−1^ associated with OH groups. The FT-IR spectra of polarized β-TCP closely resembled that of unpolarized β-TCP.

### 2.3. Osteoblast Proliferation Assay

The CCK-8 assay was used to measure MC3T3-E1 proliferation (Figure 4). Optical density (OD) values increased over time in all groups. On the first day, no significant differences were observed between the groups. By Day 3, the OD values of the positively charged β-TCP (P-β-TCP) were significantly higher than those of the β-TCP (* *p* < 0.05). On day 7, both the negatively charged (N-β-TCP) and P-β-TCP samples showed significant increases in their OD values compared with the β-TCP group (* *p* < 0.05).

### 2.4. Osteoblast Differentiation Assay

To evaluate the effects of electrical polarization on cell differentiation, alkaline phosphatase (ALP) levels were measured in MC3T3-E1 cells cultured on β-TCP, P-β-TCP, and N-β-TCP samples after 7, 11, and 14 days (Figure 5). ALP levels increased over the 14-day culture period in all specimens. After day 7, the P-β-TCP group exhibited significantly higher ALP activity than the β-TCP group (* *p* < 0.05). By day 14, both the P-β-TCP and N-β-TCP groups demonstrated significantly higher ALP activity compared with the β-TCP group (* *p* < 0.05).

### 2.5. TRAP Staining and TRAP Activities

To investigate the influence of surface charge on osteoclast differentiation, TRAP staining was conducted after incubation for 3 and 5 days in differentiation media. TRAP-positive osteoclasts were observed in all samples after 3 days, with a higher prevalence in the N-β-TCP group than those in the other groups (Figure 6c).

This observation was corroborated by TRAP activity assays (Figure 6g), which showed significantly higher enzymatic activity in the N-β-TCP group than those in the β-TCP and P-β-TCP groups on day 3. Over time, the number of TRAP-positive osteoclasts increased across all groups, with both uncharged and charged β-TCP surfaces showing an overall increase in the TRAP activity by day 5. Notably, the β-TCP group exhibited significantly lower TRAP activity compared with both the N-β-TCP and P-β-TCP groups (* *p* < 0.05), while no statistically significant difference was observed between the P-β-TCP and N-β-TCP groups on day 5. Additionally, the cells cultured on the charged β-TCP surfaces tended to be larger and multinucleated, whereas those cultured in the β-TCP groups were predominantly mononuclear, with fewer distinct multinucleated cells (Figure 6).

### 2.6. β-TCP Resorption Capacity

The absorptive capacity of the derived osteoclasts was quantified by seeding the samples with osteoclasts and measuring the volume of the pits. Compared with the small circular pits formed in the β-TCP group, the pits in the N-β-TCP group were relatively larger and deeper, with diameters of ~300 μm (Figure 7c,f), whereas the pits in the P-β-TCP group were numerous and irregular (Figure 7b,e). A statistical comparison revealed that the mean pit volume induced by β-TCP stimulation was significantly higher in the presence of a surface charge than that in the uncharged group (* *p* < 0.05).

## 3. Discussion

The success of bone substitute materials depends on their structural and chemical properties and interaction with the host tissues. This study was aimed at examining the in vitro biological interactions between pure and surface-charged β-TCP using cultured osteoblasts and osteoclasts.

The polarization method used in this study shows potential for modifying bioceramic surfaces to enhance their biological performance. Current research indicates that the cellular behavior of implanted biomaterials can be manipulated by polarization-induced surface charges on these biomaterials, which involves the adsorption of inorganic ions, carbohydrates, and various proteins, thus influencing tissue and organ regeneration [39,40,41]. Polarized HAp exhibits increased surface hydrophilicity, which is advantageous for the adhesion and proliferation of osteoblasts [32,42]. Progress has been made toward elucidating the polarization mechanism. To date, there is a lack of commercially available synthetic grafts or implants with artificially designed surface charges, mainly due to a limited understanding of the role of surface charges in biological activities [43]. A method has been developed for controlling the surface charges on β-TCP samples via polarization, offering new insights into the future research and development of functionally enhanced β-TCP scaffolds and implants.

The rationale for selecting β-TCP samples in this study for polarization research is derived from the clinical perspective of developing implants suitable for addressing extensive bone defects. Previous in vivo studies were primarily focused on β-TCP in a particulate form, and the biological effects of polarized β-TCP with different surface charges on osteoblasts and osteoclasts have not been sufficiently elucidated [35,37]. Moreover, the implants required for addressing substantial bone defects in clinical settings are often large and necessitate personalized customization based on the shape of the defect site [44,45]. In this study, the effects of surface charges on β-TCP with different polarizations on osteoblasts and osteoclasts were compared, with the aim of revealing the influence of polarization on the biological activity of β-TCP at the cellular molecular level.

An XRD analysis primarily revealed the crystal structure of the substance, while FT-IR spectroscopy was mainly employed for elucidating surface chemical properties. Neither the XRD patterns of β-TCP nor its FT-IR spectra showed any significant differences before and after polarization, indicating that polarization does not affect its crystal structure and surface chemical properties. According to past research, the HAp surface is composed of a single-phase hexagonal HAp both before and after polarization, with no significant change in elemental composition, as revealed by an XPS analysis [46]. This suggests that polarization may enhance the surface charge storage capacity without altering the surface chemical properties of the biomaterial.

Cell growth is a fundamental biological behavior involved in tissue reconstruction and organ regeneration [47,48]. Excellent biocompatibility is a key factor in ensuring the successful application of artificial bone materials [49]. The biocompatibilities of both surface-charged and uncharged β-TCP were investigated using the CCK-8 assay with osteoblast MC3T3-E1 cells. From day 3 onward, cell proliferation increased on the P-β-TCP, and by day 7, cell proliferation on the charged β-TCP surface was higher than that in the uncharged group, regardless of the charge. These results indicated that polarization is conducive to osteoblast proliferation on β-TCP. ALP activity is commonly utilized as an early indicator of bone formation [50]. Its measurement can be employed to assess successful differentiation of cells toward osteoblasts under culturing conditions. In cell differentiation experiments, from day 11 onward, the level of ALP secretion by cells on the P-β-TCP was significantly different compared with that of the β-TCP group. After 14 days, the ALP levels in both the P-β-TCP and N-β-TCP groups were significantly different from that in the β-TCP group (* *p* < 0.05). MC3T3-E1 cells exhibited similar trends in proliferation and differentiation across different experimental groups.

On negatively charged surfaces, polarization has been demonstrated to enhance bone-like hydroxyapatite deposition [51] and increase osteoblast adhesion, spreading, proliferation, or extracellular matrix deposition [30,52]. Kizuki et al. also reported that MC3T3-E1 cells were more abundant on positively charged surfaces [53]. Other studies have also affirmed the positive effects of polarization on the adhesion, attachment, and proliferation of osteoblast-like cells [54,55].

Osseointegration is governed by multiple factors, such as host cells, implants, and cytokines, which collectively contribute to bone reconstruction. Both osteoclasts and osteoblasts are crucial for successful osseointegration [56,57]; osteoblasts play a key role in bone formation, maintenance, and remodeling [58,59,60], while osteoclasts mainly regulate the resorption of mineralized tissues or materials, which is an essential aspect of osseointegration [61,62,63]. Osteoclast-induced bone resorption is coordinated with osteoblast-mediated bone formation. After implantation, osteoclast-mediated resorption is initiated around the implant, which is accompanied by bleeding and inflammation [62]. Osteoclasts are also involved in bone formation, because prior studies suggest that they resorb the bone matrix and release active transforming growth factor-β1 (TGF-β1), recruiting MSCs to stimulate bone formation [64]. Therefore, osteoclast activity should be considered when promoting bone regeneration in bone defects.

Polarized CA can enhance osteoclast resorption without affecting TRAP staining or morphology [65]. In this study, TRAP staining studies (Figure 6) revealed that negatively charged β-TCP surfaces exhibited consistently higher TRAP activity, suggesting enhanced osteoclast activity with greater potential for bone resorption. This observation aligns with previous reports that the charge on a bioceramic material surface promotes osteoclast activation [65]. Additionally, elevated TRAP activities on days 3 and 5 implied a sustained regulatory effect of the surface charge on osteoclast function. Surface-charged β-TCP exhibited superior osseointegration and significantly induced osteoclastogenesis, with enhanced activity favoring osseointegration over uncharged surfaces. The TRAP staining results suggest that the surface charge significantly influenced the morphology and size of RAW264.7 cells on β-TCP. Osteoclasts on charged β-TCP appeared larger with more nuclei, adopting round and oval shapes, while those on uncharged β-TCP were smaller and irregular with fewer nuclei (Figure 6).

In this study, we compared the effects of osteoclasts on the resorption of β-TCP with and without a surface charge and found significant differences between the groups. Scanning electron microscopy (SEM) images showed that osteoclasts on N-β-TCP surfaces produced larger and deeper resorption pits (Figure 7c,f) with diameters of ~300 µm. On P-β-TCP surfaces, pits were numerous and irregular with diameters of ~100 µm. Fewer and smaller pits were observed on β-TCP. Using 3D laser microscopy (LEXT OLS4100, Olympus, Tokyo, Japan), resorption pits were observed on all sample surfaces. Measurements indicated that the osteoclast resorption volume on charged samples was approximately double that on uncharged samples (Figure 7g). Osteoclasts play a critical role in the resorption of bone minerals through acidification, which facilitates their dissolution. Similarly, the degradation of calcium phosphate bioceramic is largely governed by osteoclast activity. These cells adhere to a bioceramic surface via the formation of a sealing zone, within which they secrete hydrogen ions (H^+^). This localized acidification reduces the pH within the sealing zone to approximately 4–5, effectively dissolving the calcium phosphate material [66,67,68]. Multinucleated osteoclasts resorb the calcium–calcified matrix to maintain a continuous remodeling process in bones [69]. Additionally, macrophages can phagocytose small β-TCP particles in vivo [70,71]. Past research has shown that the macrophage phagocytosis of particles stimulates bone resorption [72]. Increased surface free energy also contributes toward enhancing osteoclast resorption on polarized CA [65]. The biodegradability of surface-charged β-TCP was superior to that of β-TCP, with distinct advantages in clinical applications, including accelerated absorption and degradation, enhanced early osseointegration, and improved wound healing.

This study demonstrates that surface charges significantly influence osteoclast formation and function, as reflected in osteoblast proliferation and differentiation, osteoclast morphology, TRAP activity, and resorption pit formation. Given the complementary roles of osteoclast resorption and osteoblast bone secretion in bone healing and recognizing the high bioactivity and solubility of β-TCP, our findings suggest that surface-charged β-TCP not only promotes osteoclast proliferation and differentiation but also enhances osteoclast activation and resorption. These findings highlight the complementary roles of osteoclast-mediated resorption and osteoblast-mediated bone deposition in bone remodeling and underscore the potential of polarized β-TCP as a material that enhances this balance. Given the inherent bioactivity and solubility of β-TCP, the surface-charged variant not only promotes osteoclast proliferation and differentiation but also enhances their activation and resorptive capacity, thereby addressing critical challenges in bone repair by facilitating balanced bone remodeling—an essential criterion for clinical efficacy. Polarized β-TCP presents significant advantages over its non-polarized counterpart, particularly in terms of bioactivity enhancement owing to surface charge modulation.

Clinically, polarized β-TCP offers the potential for accelerated integration with host tissues and shortened healing times, making it a promising candidate for addressing complex bone defects. However, several challenges remain, including the long-term stability of surface polarization effects in vivo, the scalability of its fabrication process, and cost considerations for widespread clinical application. Furthermore, the immune response to polarized β-TCP and its degradation profile under physiological conditions require comprehensive evaluation. Future research should focus on resolving these issues while optimizing the unique properties of the material to facilitate the transition from experimental studies to clinical implementation.

## 4. Materials and Methods

### 4.1. Preparation of β-TCP Samples

The β-TCP powder was synthesized using a solid-state reaction method at a specific Ca/P molar ratio of 1.50 with CaCO_3_ and NH_4_H_2_PO_4_ (FUJIFILM Wako Pure Chemicals Corporation, Osaka, Japan), following a procedure outlined in previous investigations [36,37,73]. Mixtures were milled in ethanol for 48 h, followed by ethanol removal via a rotary evaporator (N-1300, Eyela, Tokyo, Japan). Subsequently, the materials were calcined at 900 °C for 24 h. After calcining the powder, it was crushed and sieved through a mesh with a size range of 75–149 μm. After uniaxial pressing (pellet dimensions: 15 mmφ × 1.2 mmt) at 110 MPa for 1 min and cold isostatic pressing (CIP: LCP-80-200A, NPA System, Saitama, Japan) at 200 MPa for 10 min, the samples were sintered at 1100 °C for 12 h.

### 4.2. Electrical Polarization and TSDC Measurement of β-TCP

According to prior studies that have documented electrical polarization methodologies [37,74], a direct current (DC) field of 0.65 V mm^−1^ was applied for 1 h at 400 °C in an air environment. The samples were cooled to room temperature under electric field polarization conditions. TSDC measurements were used to determine the charges on the surfaces of polarized β-TCP samples (Figure 1). The samples were clamped between platinum electrodes and were heated at a ramp rate of 5 °C min^−1^, reaching approximately 650 °C. Measurements were conducted using a galvanometer (6514/J, Tektronix Inc., Tokyo, Japan). The stored charges in the polarized β-TCP samples were determined as follows:(1)$$Q=\frac{1}{\beta }\int J\left(T\right)dT$$

Here, *Q* is the stored charge (C cm^−2^), *β* is the temperature ramp rate (K s*^−^*^1^), and *J* (T) is the depolarization current density (A cm*^−^*^2^).

### 4.3. Surface Characteristics

The crystal structure of the β-TCP samples was determined by an XRD analysis using a Mini Flex 600 instrument (Rigaku Corp., Tokyo, Japan) to carry out Bragg–Brentano geometry with a fast 1D detector (D/teX Ultra) employing Cu-Kα radiation (40 kV, 15 mA). This analysis was conducted over the diffraction angle range of 10–60° (2θ) at a scan speed of 10° min*^−^*^1^.

Infrared absorption spectra of the samples were acquired by the KBr method using a Fourier-transform infrared spectrophotometer (FT-IR-4200, JASCO Corp., Tokyo, Japan) within a spectral range of 400–4000 cm*^−^*^1^.

### 4.4. Cell Culture

The MC3T3-E1 osteoblast-like cell line and RAW264.7 osteoclast-like cells were used in this study and cultured in Alpha-modified Eagle’s medium (α-MEM), and then were added to 10% fetal bovine serum (FBS) and 1% penicillin/streptomycin at 37 °C and 5% CO_2_. Every three days, the culture medium was changed. For biological experiments, MC3T3-E1 cells were detached using trypsinization, suspended in a new medium, and used according to the experiment design. The RAW264.7 cells were removed from the flask substrate, the suspension was aspirated, and appropriate amounts were added to new culture vessels.

### 4.5. Osteoblast-like Cell Proliferation

The MC3T3-E1 proliferation rate was assessed using the Cell Counting Kit-8 (DOJINDO Laboratories, Kumamoto, Japan) (CCK-8). Briefly, a cell density of 5 × 10^4^ cells mL*^−^*^1^ was seeded on β-TCP, P-β-TCP, and N-β-TCP samples. Incubation was then conducted for 1, 3, and 7 days, followed by further incubation at 37 °C for 2 h with a CCK-8 solution added to the wells. Subsequently, an absorbance microplate reader at 450 nm was used to measure the absorbance of each well.

### 4.6. Osteoblast-like Cell Differentiation

The cells were cultured with samples at a density of 5 × 10^4^ cells mL*^−^*^1^, and the medium was substituted with an osteogenic culture medium comprising α-MEM with 10 mM β-glycerophosphate, 0.1 µM dexamethasone, and 50 μg mL*^−^*^1^ ascorbic acid after 2–3 days. ALP activity was assessed on days 7, 11, and 14, and p-nitrophenol (p-NP) was quantified by quantifying its release from p-nitrophenyl phosphate (p-NPP). The MC3T3-E1-seeded samples were gently rinsed thrice with PBS. The cells were lysed for 30 min in 40 µL of PBS containing 0.5% Triton X-100. For optimal cell lysis, three cycles of freezing and thawing were employed. The lysate was then incubated with a p-NPP solution for 15 min at 37 °C. A stop solution of 80 μL was added to terminate the reaction. Microplate readers were used to determine the ALP activity by measuring the absorbance at 405 nm.

### 4.7. Tartrate-Resistant Acid Phosphatase (TRAP) Activity Assay

The activity of TRAP was quantified using the TRAP Quantification Kit (TAKARA MK301, Takara Bio Inc., Shiga, Japan). Seeded RAW264.7 cells were arranged at a density of 2.5 × 10^4^ cells cm*^−^*^2^. After 24 h, the medium was replaced by an osteoclastogenic medium, and the cells were incubated for two days at 37 °C, with the medium being changed every two days. On days 3 and 5, the supernatants were removed, and the samples were washed with saline. Subsequently, 20 µL of cell extraction solution was added to each well and mixed by gentle pipetting. The enzymatic reaction was initiated by adding 50 µL of the reaction substrate solution, followed by 30 min of incubation at 37 °C. Using a microplate reader(iMark microplate reader, Bio-rad Laboratories, Inc., Hercules, CA, USA), the absorbance at 405 nm was determined after stopping the reaction with 50 µL of 0.5 N NaOH.

### 4.8. Osteoclast Formation Assay

Osteoclast formation was assessed using a TRAP staining kit (TAKARA MK300, Takara Bio Inc.). On days 3 and 5, the cells were washed with a sterile PBS solution after removing the culture supernatant. After fixing the cells for 5 min with a fixation solution at room temperature, they were washed with sterile distilled water (DW). The acid phosphatase substrate solution was added to the wells, and the reaction was carried out at 37 °C for 30 min. After the reaction, the samples were washed thrice with sterile DW. TRAP-positive cells were identified and quantified under a microscope.

### 4.9. Visualization and Measurement of Resorption Pit

On day 10, the samples were treated for 10 min with a 0.5% Triton X-100 solution. After confirming complete cell dissolution under the microscope, each sample was rinsed thrice with PBS solution. Pit formation was examined under a scanning electron microscope (JSM-7900F, JEOL, Tokyo, Japan). CLSM imaging was performed using a 3D measurement laser microscope (LEXT OLS4000, Olympus, Tokyo, Japan). The area and volume of the resorption pits were measured using a dedicated software (OLS4000 2.1). Quantification of the resorbed areas was then performed using the particle analysis function. Five regions were analyzed per sample, with three samples per group.

### 4.10. Statistical Analysis

In all cases, the data are expressed as mean ±  standard deviation, based on the results obtained from a minimum of three independent experiments. Statistical analysis was conducted using the GraphPad Prism version 10 (GraphPad, San Diego, CA, USA) and employing one-way analysis of variance (ANOVA) and two-way ANOVA, followed by Tukey’s multiple comparison test. A *p*-value of 0.05 was determined to be statistically significant.

## 5. Conclusions

In this study, β-TCP materials with surface charges were successfully developed, exhibiting physicochemical properties comparable to those of currently utilized clinical materials while offering enhanced biocompatibility. The experimental results from osteoblast and osteoclast cultures demonstrated that polarized β-TCP materials significantly enhanced osteoblast proliferation and differentiation. Furthermore, they effectively promoted osteoclast activity and resorption, indicating their potential to improve bone regeneration and integration in clinical applications.

## Data Availability

The data presented in this study are available on request from the corresponding author.

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
