# Peer review of "Exploring the Biological Impact of β-TCP Surface Polarization on Osteoblast and Osteoclast Activity"

_ijms, 2024, doi:10.3390/ijms26010141_

Round 1

Reviewer 1 Report

Comments and Suggestions for Authors

This manuscript investigates the effects of β-TCP surface polarization on osteoblast proliferation, differentiation, and osteoclast activity. The authors hypothesize that β-TCP with a highly dense surface charge would promote osteoblast proliferation and differentiation, while also enhancing osteoclast activity and resorption capacity. The data presented does not fully support the conclusions made regarding the effects of β-TCP surface charge on osteoblast and osteoclast activity. Additionally, the manuscript requires substantial improvements in writing clarity and the experimental methodology.

Comments on the Quality of English Language

The quality of  english requires improvement.

Reviewer 2 Report

Comments and Suggestions for Authors

This study investigates the biological impacts of β-tricalcium phosphate (β-TCP) surface polarization on osteoblast and osteoclast activity. The findings reveal that polarized β-TCP enhances osteoblast proliferation and differentiation, as well as osteoclast activity and resorption, demonstrating its potential to improve bone regeneration and integration in clinical applications. The results are of scientific significance and biomedical relevance, making the study suitable for publication. 

Reviewer 3 Report

Comments and Suggestions for Authors

The topic of this paper is very interesting and it will be of interest to the readers of the IJMS. The paper is in general well-written; conducted methods are described in detail and obtained results are clearly presented.

However, there are some issues in the Introduction, Discussion and Figures that should be addressed prior publication. 

1. The Introduction should be expanded and describe in more detail state of the art in the field. Specifically, the authors should describe clinical application of bioceramics and novel trends in the development of bioceramic materials (surface modifications, combinations with growth factors-such as BMP or VEGF).

2. The authors stated that their goal is to develop polarized TCP suitable for clinical application. Results of the present paper should be discussed in this context and authors should better perspective of polarized TCP in the bone regeneration and discuss potential advantages and disadvantages of its potential clinical application. 

3. Font sizes and style should be uniformed on all Figures (e.g. font size on Figures 1-3 is relatively small while the font size on Figures 4-5 is relatively large)

4. Scheme 1 - I would recommend to expand the scheme description

Reviewer 4 Report

Comments and Suggestions for Authors

The document titled “Exploring the Biological Impact of β-TCP Surface Polarization on Osteoblast and Osteoclast Activity” presents an exhaustive study focused on bone regeneration, utilizing innovative techniques to enhance materials, particularly through the application of electrical cues that significantly affect their properties and the interaction of cells with the material. The results of this research could provide substantial value to the field. However, I have outlined several recommendations aimed at improving the document:

-The quality of the images needs improvement, as they currently appear unclear.

- Please enhance Figure 6 by adding the corresponding sample labels for better clarity.

- Figures A and D are not referenced in the main text; please ensure all figures are appropriately cited.

- What additional studies could be conducted to elucidate the effect of polarization on the HPA material? It is evident that the techniques employed did not demonstrate significant changes in the material.

- Why did the cells exhibit a better response to positively charged materials compared to negatively charged and uncharged materials?

- Is the change in cell morphology induced by the material's charge beneficial? What potential effects could this have on bone regeneration?

Round 2

Reviewer 1 Report

Comments and Suggestions for Authors

Thank you for revising the draft in response to the previous comments. While the manuscript shows improvement, it still lacks sufficient background literature regarding the use of hydroxyapatite nanoparticles in bone-related applications. To strengthen the introduction and provide a more comprehensive context for your work, I recommend discussing relevant studies that highlight the role of hydroxyapatite nanoparticles in bone regeneration, osteoconductivity, and their interaction with biological systems.

Here are some examples of relevant research articles: doi: 10.1016/j.msec.2016.02.062; doi: 10.1021/acsami.8b02792.]; doi: 10.1016/j.msec.2016.10.066;doi: 10.1016/j.ejps.2014.10.015.
